# A Retrospective Cohort Study of Cobra Envenomation: Clinical Characteristics, Treatments, and Outcomes

**DOI:** 10.3390/toxins15070468

**Published:** 2023-07-20

**Authors:** Phantakan Tansuwannarat, Achara Tongpoo, Suraphong Phongsawad, Charuwan Sriapha, Winai Wananukul, Satariya Trakulsrichai

**Affiliations:** 1Chakri Naruebodindra Medical Institute, Faculty of Medicine Ramathibodi Hospital, Mahidol University, Bangkok 10540, Thailand; phantakan.tans@gmail.com (P.T.); s.phongsawad@gmail.com (S.P.); 2Ramathibodi Poison Center, Faculty of Medicine Ramathibodi Hospital, Mahidol University, Bangkok 10400, Thailand; achara.ton@mahidol.ac.th (A.T.); charuwan.sri@mahidol.ac.th (C.S.); winai.wan@mahidol.edu (W.W.); 3Department of Medicine, Faculty of Medicine Ramathibodi Hospital, Mahidol University, Bangkok 10400, Thailand; 4Department of Emergency Medicine, Faculty of Medicine Ramathibodi Hospital, Mahidol University, Bangkok 10400, Thailand

**Keywords:** *Naja kaouthia*, *Naja siamensis*, *Naja sumatrana*, monocled cobra, Indochinese spitting cobra, Equatorial spitting cobra, golden spitting cobra, deaths

## Abstract

This study investigated the clinical characteristics, treatments, and outcomes of envenomation involving cobra species in Thailand (*Naja kaouthia*, *Naja siamensis*, and *Naja sumatrana*). Data of patients who had been bitten by a cobra or inoculated via the eyes/skin in 2018–2021 were obtained from the Ramathibodi Poison Center. There were 1045 patients admitted during the 4-year study period (bite, n = 539; ocular/dermal inoculation, n = 506). Almost all patients with ocular/dermal inoculation had eye involvement and ocular injuries, but none had neurological effects. Most of the patients bitten by a cobra had local effects (69.0%) and neurological signs and symptoms (55.7%). The median interval between the bite and the onset of neurological symptoms was 1 h (range, 10 min to 24 h). Accordingly, patients should be observed closely in hospitals for at least 24 h after a bite. Intubation with ventilator support was required in 45.5% of patients and for a median duration of 1.1 days. Antivenom was administered in 63.5% of cases. There were nine deaths, most of which resulted from severe infection. Neurological effects and intubation were significantly more common after a monocled cobra bite than after a spitting cobra bite. The administration of antivenom with good supportive care, including the appropriate management of complications, especially wound infection, might decrease fatality.

## 1. Introduction

Snakebite envenomation is a neglected tropical condition that is estimated to affect millions of people and is associated with considerable morbidity, disability, and mortality worldwide [1,2]. The highest envenomation rates are reported in South Asia, followed by Southeast Asia and East Sub-Saharan Africa [2]. Southeast Asia is home to numerous species of venomous snake and is among the tropical regions with a high incidence of snakebite envenomation and deaths annually [1,2,3,4].

In Thailand, snakebite envenomation is a commonly encountered medical emergency and public health problem that is associated with considerable morbidity and mortality [5,6,7,8,9,10,11,12,13,14,15]. Venomous snakes in Thailand comprise four species in the genus *Naja* [1,16]. *Naja kaouthia* (the monocled cobra) and *Naja siamensis* (the Indochinese spitting cobra) are listed in category 1 (of the highest medical importance) in the World Health Organization classification of venomous snakes [1,16]. The other two species are *Naja sumatrana* (the Equatorial or golden spitting cobra) [1,16] and *Naja fuxi*, which is a newly identified species in Thailand [17,18].

A cobra bite can have local effects and cause post-synaptic neurotoxicity by blocking transmission at the neuromuscular junction, which has neurological effects, including paralysis [1,19].

Neurological envenomation includes ptosis, ophthalmoplegia, paralysis of the facial muscles and other muscles innervated by the cranial nerves, a nasal voice, bulbar paralysis, difficulty swallowing, and respiratory and generalized flaccid paralysis that can lead to death in patients with severe envenomation [1,10,12,15,19].

Although cobra envenomation has traditionally been a problem concentrated in Asia, cobra bites are now being reported in other regions and may reflect an increase in trade in exotic and non-native venomous snakes [20,21,22,23,24]. For example, the monocled cobra (*N. kaouthia*) is now a common species and one of the causes of exotic snake envenomation in the USA [20,21,23,24].

Spitting cobras are the most common cause of venom ophthalmia [25,26,27,28]. Spitting cobras in the Elapidae family spit venom from the Duvernoy’s gland, and some in the Colubridae family are able to spray a toxin from the nuchal gland [25]. *N. siamensis*, which inhabits Thailand, has been reported to be able to inoculate via the ocular route and cause eye injury [28]. Although complications, including corneal injury, conjunctivitis, keratitis, and blepharitis, can occur following direct ocular inoculation with venom, most patients recover without sequelae. However, there is limited information on systemic envenomation in patients with venom ophthalmia [25,26,27,28]. There is one reported case of systemic neurotoxicity associated with ocular contact with coral snake (*Micrurus tener*) venom [29]. The patient in this case was also bitten by the snake and eventually received Central American coral snake (*Micrurus nigrocinctus*) antivenom.

Many patients who sustain direct ocular inoculation with cobra venom or a cobra bite have a consultation and are referred to the Ramathibodi Poison Center in Bangkok, Thailand. The cobra is one of the most common venomous snakes that bite patients who attend this poison center. Cobra envenomation is relatively common and has potentially fatal complications, including muscle weakness and local necrosis, so it should not be overlooked. The Thai cobra bite studies performed to date have included only small numbers of patients [9,10,12,13,15]. Therefore, this study was performed to delineate the clinical characteristics, treatments, and outcomes of patients with cobra envenomation in Thailand.

## 2. Results

A total of 1045 cases of cobra envenomation (bite, n = 539; ocular/dermal inoculation, n = 506) were recorded during the study period. Almost all patients with ocular/dermal inoculation were inoculated via the ocular route. Four patients were inoculated via the dermal route; three of these patients were inoculated via both the eyes and skin, and one via the skin only (at the ankle).

The demographic and clinical characteristics of patients with direct ocular/dermal inoculation with cobra venom are shown in Table 1. Most were male and from the northern region. There were no patients from Bangkok. Almost all patients (92.2%) in this group sustained ophthalmic injuries. Skin irritation was not reported. Only four patients did not complain of any symptoms. None of these patients developed neurological symptoms during their hospital stay.

Two hundred and fifty-three (46.9%) of the patients with cobra bites brought the culprit snake to the hospital. Table 2 shows the demographic and clinical characteristics of this group of patients. Most patients were male (58.8%), and the median age was 43 years (range, 1–99). Forty-three patients were aged younger than 15 years. Most of the bites were on the lower extremities. The bites were discovered to have occurred in every region of Thailand. Regarding the patients with direct ocular/dermal inoculation, there were no reports of spitting cobra bites occurring in Bangkok. Fang marks were observed in almost all cases. Fang marks were not detected in two cases. Neither of the patients in these two cases developed local or systemic effects after being bitten. Most of the patients presented to our poison center during the evening. A tight tourniquet was used in only eight patients prior to arriving at the hospital.

At presentation, local effects were observed at the site of the bite wound in 372 patients (69.0%), the most common being local swelling (n = 369) and/or skin necrosis (n = 120). High blood pressure, tachycardia, tachypnea, bradycardia, and hypotension were noted at presentation in 206, 148, 100, 5 and 5 patients, respectively. No cases of hypotension were diagnosed as an anaphylactic reaction to venom.

Three patients with cobra bites presented with cardiac arrest, which was attributed to respiratory failure and apnea before arrival at the hospital. Only one of these three patients survived.

The laboratory findings at presentation are described in Table 3. The mean and median of the laboratory results were within normal range. At presentation, thirteen patients had hyponatremia (serum sodium < 135 mEq/L). Forty-four patients and two patients had hypokalemia (serum potassium < 3.5 mEq/L) and hyperkalemia (serum potassium > 5 mEq/L), respectively.

The details of patient management during hospitalization are shown in Table 4. Most patients (97.2%) were admitted to hospital.

At presentation or during hospitalization, 300 patients (55.7%) were observed to have neurological effects, which included ptosis (n = 258), muscle weakness (including limb or respiratory muscles) (n = 79), bulbar palsy (n = 54), dysphagia (n = 45), and ophthalmoplegia (n = 3). Eighty-one of the patients who developed neurological signs and symptoms did not have local effects at presentation. Necrotizing fasciitis was diagnosed in 51 patients (9.5%). The median interval between the bite and the onset of neurological symptoms was 1 h, with the longest interval being 24 h. Endotracheal intubation with ventilator support was required in 45.5% of all cases and for a median duration of 1.1 days. Most patients (63.5%) received antivenom. All patients with neurological effects received antivenom. However, some patients who did not have neurological effects, received antivenom.

One of the snakebite cases involved a 51-year-old man who was bitten on his left ankle and right index finger and admitted to a surgical ward with a gangrenous wound site that was diagnosed as necrotizing fasciitis. Fasciotomy and debridement were performed on the day following the bite. Postoperatively (about 2 days after the bite), he developed neurological symptoms of dysphagia and grade 4 motor weakness in all extremities. One dose of monovalent cobra antivenom was administered, after which the patient improved and was discharged approximately 2 weeks after admission. In view of the unusual manifestation of envenomation, this case was not included when calculating the median interval between the bite and the onset of neurological symptoms.

One case involved a 23-year-old woman who was bitten by a spitting cobra when she was 12 weeks’ pregnant. She developed local wound swelling but had no systemic effects. She was admitted, treated with antibiotics, and discharged after 1 day.

One hundred and forty-six patients received antibiotics, the most common of which were amoxicillin/clavulanic acid (n = 83), ceftriaxone and clindamycin (n = 20), and amoxicillin/clavulanic acid and clindamycin (n = 12).

Approximately 40% of patients required surgery. Fourteen patients developed complications, the most common being pneumonia (Table 4).

Nine deaths were reported during the study period (Table 5), giving an overall mortality rate of 1.7%. All deaths occurred in patients with monocled cobra bites. Three of these patients did not receive antivenom; in one case, this was because of cardiac arrest at presentation that proved fatal (n = 1), and in the other two cases, it was because of wound infection and sepsis that developed in the absence of neurological symptoms. Four of these patients were diagnosed with necrotizing fasciitis. The causes of death were reported as severe wound infection with sepsis and complications (n = 5), respiratory failure with hypoxia (n = 3), and ventilator-related complications (n = 1).

Some patients were not followed up by a specialist in poison information through to the end of their admission because their neurological symptoms had resolved, but their admission was extended for long-term wound care or for other complications that developed during hospitalization.

The clinical characteristics and treatments were compared between patients who had been bitten by a monocled cobra and those who had been bitten by two spitting cobras (Table 6). There were statistically significant between-group differences in neurological effects, including ptosis, bulbar palsy, and muscle paralysis, and in the proportion of patients who required endotracheal intubation and ventilator support. However, there were no significant differences in age, sex, local effects, abnormal vital signs at presentation, duration of intubation, surgical management, or wound management between the groups.

## 3. Discussion

This study is a large study of cobra envenomation regarding *Naja kaouthia, Naja siamensis*, and *Naja sumatrana*.

We describe the clinical characteristics, treatments, and outcomes in patients who have been bitten by a cobra or directly inoculated with cobra venom via the eyes or skin in Thailand.

A previous retrospective study of RPC data that identified 1006 envenomed patients over a 2-year period [30] found cobra bites (n = 271) to be the most common type of snakebite in patients with neurotoxic envenomation (n = 356). Antivenom was administered in 188 of these cases.

In this study, almost all patients with direct inoculation of cobra venom via the eyes and/or skin developed ocular symptoms. However, there were no severe symptoms, such as permanent loss of vision. All patients received only supportive and symptomatic treatments, such as eye irrigation, and were discharged with good outcomes. Our finding that the main clinical effects of venom ophthalmia caused by snake venom are ocular irritation, inflammation, and injury is concordant with previous reports [25,28]. However, in line with our routine practice, we did not follow up with these patients after discharge from hospital, so whether there were any persistent signs or symptoms or further effects beyond the clinical findings recorded during the hospital stay is unknown. Nevertheless, there were no records concerning repeat consultations for worsening ocular effects after discharge from hospital. There was a case report of systemic envenomation that manifested as neurotoxicity in a patient exposed to *M. tener* venom via the ocular mucus membranes together with a small cutaneous bite [29]. However, there were no reports of systemic envenomation in our study. Our results indicate that ocular inoculation by either of the two spitting cobra species included in this study caused mainly mild eye irritation with no systemic effects and did not require specific treatment, such as the administration of antivenom.

Our finding that most of the patients who presented with a cobra bite were male and bitten on a lower extremity is similar to the findings of previous studies of cobra bites in Thailand [10,15,31]. In terms of epidemiological distribution, cobra bites occur in every region of Thailand, especially in the central and northeastern parts of the country. However, there was a difference in the distribution of patients bitten by the monocled cobra and those bitten by either of the two spitting cobra species, as shown in Table 2. Eye and/or skin inoculation and bites by spitting cobras were not reported in Bangkok. Bites by the monocled cobra were uncommon in the northern region and bites by either of the spitting cobras were uncommon in the western region. These epidemiological data might help to identify the culprit species of cobra, although patient management is similar regardless of species.

Local effects such as swelling or necrosis were common in our patients, as in previous reports [10,15,31,32]. Some patients presented with high blood pressure, tachycardia, and tachypnea, and a smaller number presented with bradycardia and hypotension. A cobra bite might have cardiovascular effects. A study in Thai children described a pulse rate of >100 beats/min in 29 patients (61.7%) and slightly elevated blood pressure in eight patients (17.0%) on admission [15]. Whether high blood pressure and/or tachycardia might be explained by the patient’s psychological distress following a snakebite or by other effects of snake venom needs further study. One case report [33] mentioned transient gastrointestinal symptoms, hypotension, and tachycardia following a cobra bite. The patient’s electrocardiogram showed ventricular bigeminy. However, the patient survived and was discharged. Snake venom contains a number of proteins and peptides, such as phospholipase A, that exert hypotensive effects via various mechanisms [34]. Cardiovascular effects were described in most patients in an Indian study of snake envenomation [35]. 

Systemic neurological effects, the most common being ptosis, were observed in approximately half of our patients. This finding is close to that in a previous study [32], lower than that in another study [15], and higher than that in a further study [10]. Ophthalmoplegia was rarely reported in our study, this might be partially attributed to under-reporting by the attending physicians, who potentially overlooked or did not notice this abnormality.

The median interval between the snakebite and the onset of neurological signs and symptoms was quite short in our study, but similar to the findings of others [15,31]. One study in children with neurotoxic snakebites [15] found that the latest neurological sign developed 19 h after the bite. The longest interval between the bite and onset of neurological symptoms in our patients was 24 h. Some patients did not have a local effect but developed neurological signs and symptoms. Therefore, close observation, especially for respiratory and neurological effects, is warranted for at least 24 h after the bite, regardless of the severity of the local effect.

Interestingly, one patient developed neurological effects after fasciotomy and debridement performed about 2 days after the bite, possibly because venom remaining at the bite wound site was released and reached the circulation after the wound surgery [7].

A tight tourniquet is not recommended in the treatment of snakebites [1] and was applied in only eight patients in our study, indicating that tight tourniquets are not used routinely in the Thai population. However, education regarding prehospital care and antivenom treatment would be worthwhile for both medical personnel and the general population.

Almost all patients with neurological effects in our study required ventilator support for a median of 1.1 days. All patients with neurological effects received antivenom regardless of whether they were bitten by a monocled or spitting cobra, in accordance with the recommendation of the Queen Saovabha Memorial Institute (QSMI) [36] and the Thai Society of Clinical Toxicology [37] that the indication for antivenom include any clinical sign or symptom of muscle weakness. However, some patients without neurological effects also received antivenom. One study found that the Thai antivenom prepared against N. *kaouthia* venom had a neutralizing effect on spitting cobra (*N. siamensis*) venom [38].

The mortality rate in our study was close to that in another study (1.2%) [10] but lower than that in two other reports (4.3% and 11.4%) [15,32]. These inconsistent findings may reflect differences in management over time or in the numbers of patients enrolled in the various studies.

The mortality rate was low in our study, but deaths from a cobra bite still occurred. The main causes of death were severe infection of the bite wound and respiratory failure as a consequence of muscle paralysis. Most of the deceased patients had received cobra antivenom but still died because of delays in airway management or severe sepsis subsequent to wound infection. One study preformed in Thailand [32] reported four deaths following cobra bites, of which three patients died from severe infection. We found that approximately 10% of all patients developed necrotizing fasciitis. Therefore, close monitoring, appropriate respiratory and wound care, and aggressive management of infection and complications are required in all patients who present with a cobra bite. A few patients in our study presented to hospital with respiratory failure and cardiac arrest. These data might facilitate the establishment of education programs for the general population regarding early transportation to hospital via the ambulance system.

We found significant differences in neurological effects and the requirement for endotracheal intubation with ventilator support between patients who had been bitten by a monocled cobra and those who had been bitten by a spitting cobra. All deaths were due to a monocled cobra bite. An in vitro study found that the neurotoxic and myotoxic effects of *N. kaouthia* venom were more potent than those of *N. sumatrana* venom [39], which is consistent with our finding that neurological effects were more common in patients bitten by a monocled cobra than in those bitten by either of the spitting cobras. Our data suggest that patients who had been bitten by a cobra, whether of the monocled or spitting variety, should be monitored closely for neurological effects and respiratory symptoms, with prompt management should they occur.

Our study has several limitations. First, the reporting of cobra bites and direct ocular and/or skin inoculation with cobra venom to the RPC is not mandatory. Therefore, not all cobra envenomation cases would have been referred to our institution during the study period, and it is possible that the true incidence of cobra envenomation and its mortality rate may be higher than reported here. Second, the study was performed retrospectively, so we cannot exclude the possibility of missing or incomplete data. Third, in line with practice at our poison center, some patients were followed up via telephone until the resolution of their clinical symptoms of envenomation but were not followed up until hospital discharge or death in hospital. Furthermore, our specialist in poison information and staff did not follow up on each patient’s clinical status after discharge from hospital. Therefore, we could not investigate long-term outcomes. Finally, the diagnosis of cobra envenomation was based mainly on patients’ reports of being bitten or inoculated by a cobra and supported by other information, including the snake’s area of distribution, the patient’s local and systemic clinical features of envenomation, laboratory abnormalities, and the response to specific antivenoms. No definitive laboratory test could be performed for a cobra bite, and blood or urine could not be analyzed for cobra venom to make a definitive diagnosis.

## 4. Conclusions

In this study, cobra spit ophthalmia did not have neurological effects. Abnormal vital signs were noted in patients with a cobra bite at presentation. The mortality rate was low at 1.7%, and all deaths occurred in patients with a cobra bite. Patients who had been bitten by a monocled cobra were more likely to have neurological effects and require endotracheal intubation with ventilator support than those who had been bitten by a spitting cobra. Patients who present with a cobra bite should be observed for at least 24 h after the bite. Adequate supportive care including the management of complications (especially wound infection), in addition to the administration of antivenom, might help to decrease the mortality rate.

## 5. Methods

### 5.1. Study Design

This study had a retrospective cohort design and analyzed data entered into the Ramathibodi Poison Center (RPC) Toxic Exposure Surveillance System database for patients who had been bitten by a cobra (either a monocled cobra or a spitting cobra) or directly inoculated with cobra venom via the eyes or skin between 2018 and 2021.

The primary outcomes were the clinical characteristics, treatments, and outcomes of cobra envenomation in Thailand. The secondary outcomes were the differences in the clinical characteristics and outcomes between patients bitten by the monocled cobra and those bitten by the spitting cobra.

This study was approved by the institutional ethics committee of Ramathibodi Hospital Faculty of Medicine, Mahidol University (COA. MURA2020/838). The requirement for informed consent was waived in view of the retrospective observational design of the research and the anonymized reporting of the confidential data obtained from the poison center database.

### 5.2. Study Site and Population

The study setting was the RPC, which is based in a tertiary teaching hospital in Thailand that serves the whole country with a 24-h telephone service for both healthcare personnel and the general public. There are approximately 25,000–30,000 consultations annually, and most queries to the center are from medical personnel. Follow-up calls are made to collect patient data and information on progress, to recommend treatment, and to ascertain medical outcomes. All cases are recorded in the RPC Toxic Exposure Surveillance System database. All records, particularly regarding diagnoses and severity, are reviewed and verified by a team of senior specialists in poison information and medical toxicologists.

All patients who had been bitten by a cobra or directly inoculated with cobra venom via the eyes or skin for whom there was a consultation with the poison center during the 4-year study period were enrolled. The cobra was determined using any of the following methods: identification of the snake or snake carcass brought to the hospital by the patient or a witness; a photograph of the snake taken by the patient or a witness; description of the morphology or species of snake by the patient or a witness, together with the area of distribution/habitat of that snake species, and/or the patient’s clinical features of neurotoxic envenomation. Confirmation of the cobra species using a snake carcass and a photograph of the snake was conducted by the poison center’s consultant (an experienced veterinarian from the Snake Farm at the QSMI, Thai Red Cross, Bangkok, Thailand). Patients known to have ingested ethanol, herbs, illicit drugs, pesticides, or other chemicals at the time of envenomation were excluded.

### 5.3. Study Protocol

Information was collected on each patient’s demographic characteristics, medical history, laboratory findings, treatments received, follow-up details, final diagnosis, and outcome.

Abnormal vital signs were defined in both adults and children according to age group [40]. Acute kidney injury was identified by the Kidney Disease: Improving Global Outcomes clinical practice guidelines [41]. We assumed that all patients without known underlying diseases had normal kidney function before snake envenomation. Rhabdomyolysis was noted if the patient’s serum creatine phosphokinase level was >1000 U/L [42] or this condition was recorded in the patient’s data.

The patients who had been bitten by a cobra were treated with monovalent or polyvalent snake antivenom produced by the QSMI. Antivenom therapy is recommended for systemic envenomation following the recommendations of the QSMI [36] and the Thai Society of Clinical Toxicology [37]. The indications for the administration of antivenom for cobra envenomation include any clinical symptom or sign of muscle weakness. Early adverse reactions to antivenom include unwanted manifestations that occur within 24 h of administration [43].

### 5.4. Statistical Analysis

The study data were collected in an Excel spreadsheet (Microsoft; Redmond, WA, USA). Continuous data are summarized as the mean and standard deviation if normally distributed and as the median (range) if not. Categorical data are presented as the frequency and percentage. Continuous data were compared between groups using the Student’s *t*-test if normally distributed and the Mann–Whitney *U* test if not. Between-group differences in categorical variables were evaluated using the chi-squared test and Fisher’s exact test. All statistical analyses were performed using STATA (StataCorp, College Station, TX, USA). A *p*-value of <0.05 was considered statistically significant.

## Figures and Tables

**Table 1 toxins-15-00468-t001:** Demographic data and characteristics of all patients with direct ocular and/or dermal inoculation with cobra venom.

Characteristics	Number (%) (n = 506)
Sex	
Male	402 (79.4%)
Female	104 (20.6%)
Age; median (min–max), years	51 (1–85)
Clinical effects *	
Eye irritation	473 (93.5%)
Decreased visual acuity	49 (9.7%)
None	4 (0.8%)
Region	
Bangkok	-
Central	125 (24.7%)
North	42 (8.3%)
Northeast	180 (35.6%)
East	101 (19.9%)
West	51 (10.1%)
South	7 (1.4%)

* Some patients had >1 clinical effect.

**Table 2 toxins-15-00468-t002:** Demographic data and characteristics of all patients with cobra bites.

Characteristics	Number (%) (n = 539)
Sex	
Male	317 (58.8%)
Female	222 (41.2%)
Age; median (min–max), years	43 (1–99)
Clinical effects at presentation *	
Local swelling	369 (68.5%)
Local skin necrosis	120 (22.3%)
Region **	
Bangkok	51 (9.5%)
Central	187 (34.7%)
North	28 (5.2%)
Northeast	128 (23.7%)
East	42 (7.8%)
West	29 (5.4%)
South	74 (13.7%)
Fang mark	
Identified	525 (97.4%)
Unidentified	14 (2.6%)
Bite site	
Upper extremity	201 (37.3)
Lower extremity	332 (61.6%)
Others	6 (1.1%)
Shift of consultation to poison center	
Morning Shift time (08.00 a.m. to 04.00 p.m.)	201 (37.3%)
Evening Shift time (04.00 p.m. to 00.00 a.m.)	280 (51.9%)
Night time (00.00 a.m. to 08.00 a.m.)	58 (10.8%)
Having neurological effects at presentation or during hospitalization	300 (55.7%)
Time to onset of neurological effects after bites; median (min–max), hours ***	1 (0.16–24)

* Some patients had >1 clinical effect. ** Number of patients bitten by *Naja kaouthia* (monocled cobra): Bangkok: 51 patients; central: 174; north: 20; north-east: 105; east: 35; west: 26, south: 74. *** Data available in 272 patients.

**Table 3 toxins-15-00468-t003:** Laboratory results of all patients at presentation.

Laboratory Results (Number of Patients with Data Available)	Value
Serum sodium (mEq/L); mean ± SD (n = 155)	139.19 ± 3.96
Serum potassium (mEq/L); mean ± SD (n = 127)	3.88 ± 0.36
Serum chloride (mEq/L); mean ± SD (n = 51)	102.40 ± 5.01
Serum bicarbonate (mEq/L); mean ± SD (n = 122)	22.24 ± 6.07
Serum creatinine (mg/dL); mean ± SD (n = 117)	1.00 ± 0.45
White blood cells (per microliter); mean ± SD (n = 117)	10,820 ± 5115.56
Platelets (per microliter); median (min–max) (n = 117)	270,000 (59,000–661,000)

**Table 4 toxins-15-00468-t004:** Management of all patients with cobra bites.

Clinical Course and Treatment	Value
Endotracheal intubation with ventilator support (% of all patients)	245 (45.5%)
Duration of intubation (median, min–max), days	1.1 (0.15–22)
Antivenom administration (% of all patients)	342 (63.5%)
Monovalent antivenom	295 (86.3%)
Polyvalent neurotoxin antivenom	41 (12.0%)
Both	6 (1.7%)
Time to antivenom treatment after bites; median (min–max), hours	3.4 (0.5–240)
Antibiotics received (% of patients who received antibiotic)	146 (27.1%)
Surgical management (% of all patients)	215 (39.9%)
Debridement	196 (91.2%)
Skin graft	8 (3.7%)
Amputation	11 (5.1%)
In-hospital complications (% of all patients)	14 (2.6%)
Pneumonia	11 (78.6%)
Rhabdomyolysis	3 (21.4%)
Hospital stays (% of 529 patients who were admitted)	
<7 days	356 (67.3%)
8–14 days	117 (22.1%)
>14 days	56 (10.6%)

**Table 5 toxins-15-00468-t005:** Demographic data and characteristics of dead patients.

Characteristics	Value
Sex (Female/Male)	6/3
Age; median (min–max), years	54 (2–89)
Bite site (upper/lower extremities)	5/4
Time to onset of neurological effects after bites; median (min–max), hours	3 (0.25–13)
Duration of intubation (median, min–max), days	3.5 (1–7)
Antivenom administration *	6
Monovalent antivenom	5
Polyvalent neurotoxin antivenom	1
Time to antivenom treatment after bites; median (min–max), hours	1.17 (1–2)
In-hospital complications	3 (33.3%)
Rhabdomyolysis	1 (33.3%)
Pneumonia	2 (66.7%)
Hospital stays; median (min–max), days	5 (0.3–26)

* No patients with allergic reaction to antivenoms.

**Table 6 toxins-15-00468-t006:** Comparison of clinical manifestations and treatments between patients who had been bitten by *Naja kaouthia* (monocled cobra) and those who had been bitten by *Naja siamensis* (Indochinese spitting cobra) or *Naja sumatrana* (Equatorial spitting cobra).

Clinical Manifestations (Number of Patients with Data Available: Monocled Cobra/Spitting Cobras)	Monocled Cobra Bites	Spitting Cobra Bites	*p*-Value
Age; median (min–max), years (485/54)	42 (1–99)	49.5 (3–86)	0.233
Sex: Male	284 (58.6%)	33 (68.1%)	0.718
Local effects at presentation			
Local swelling (485/54)	330 (68.0%)	39 (72.2%)	0.531
Local necrosis (485/54)	104 (21.4%)	16 (29.6%)	0.373
Abnormal vital signs at presentation			
Hypertension (191/25)	181 (94.8%)	25 (100%)	1.000
Tachycardia (351/42)	134 (38.2%)	14 (33.3%)	0.781
Tachypnea (330/41)	91 (27.6%)	9 (21.9%)	0.250
Neurological effects			
Bulbar palsy (485/54)	54 (11.1%)	-	0.010 *
Muscle paralysis (485/54)	78 (16.1%)	1 (1.9%)	0.005 *
Ptosis (485/54)	250 (51.5%)	8 (14.8%)	<0.001 *
Endotracheal intubation with ventilator support (485/54)	235 (48.5%)	10 (18.5%)	<0.001 *
Duration of intubation (median, min–max), days (485/54)	1.65 (0.5–22)	1.1 (0.15–20)	0.733
Local effects during hospitalization			
Cellulitis (479/54)	262 (54.7%)	34 (63.0%)	0.247
Skin necrosis (476/54)	161 (33.8%)	18 (33.3%)	0.942
Necrotizing fasciitis (485/54)	47 (9.7%)	4 (7.4%)	0.587
Wound management			
Amputation (485/54)	20 (4.1%)	1 (1.9%)	0.711
Debridement (485/54)	177 (36.5%)	23 (42.6%)	0.379
Skin graft (476/54)	19 (3.9%)	-	0.242

* *p*-value < 0.05.

## Data Availability

The data are not available for public access because of patient privacy concerns but are available from the corresponding author upon reasonable request.

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
