# Peer review of "A Retrospective Cohort Study of Cobra Envenomation: Clinical Characteristics, Treatments, and Outcomes"

_toxins, 2023, doi:10.3390/toxins15070468_

Round 1

Reviewer 2 Report

Dear Respected Authors;

I reviewed your manuscript entitling: “A retrospective cohort study of cobra envenomation: clinical characteristics, treatments, and outcomes”.

I found it very interesting with scientific results.

There are minor corrections that are better to be realized before publishing:

The data in the tables, especially Table 3, can be better explained by adding a column to it.

Best Regards

Reviewer 3 Report

In RESULTS

Line 221    The clinical characteristics were compared between patients who were bitten by a monocled cobra and those who were bitten by two spitting cobras (Table 5). The correct is Is table 6

Comments: Please check carefully the reference item.........According to author guidelines, fascicle numbers are not necessary.... References should be described as follows, depending on the type of work:

 Journal Articles:
1. Author 1, A.B.; Author 2, C.D. Title of the article. Abbreviated Journal Name YearVolume, page range.

Reviewer 4 Report

This is a retrospective study of telephone consultations about cobra bites in Thailand. The study is interesting, especially for physicians in that area of Asia. The authors are aware of the study's limitations, which are well described at the end. The conclusions are correct and consistent with the presented results. However, it is not clear what is new and has not been described in the literature yet.

It is not clear from the text what neurological signs they had upon admission to the hospital, or if they had any at all, as these are only described during hospitalization. Upon admission to the hospital, only local signs, cardiovascular signs, and laboratory results are listed.

What was the time from the bite to admission to the hospital and from admission to antivenom application? Was antivenom administration delayed until the onset of neurological signs (muscle weakness)?

All columns in tables should be aligned to the left, and normal values could be described in the tables. Furthermore, antivenoms and their application could be described in more detail.
